# Plasmonic monolithic lithium niobate directional coupler switches

Martin Thomaschewski [1]*, Vladimir A. Zenin [1], Christian Wolff [1] & Sergey I. Bozhevolnyi [1]*

Lithium niobate (LN) has been the material of choice for electro-optic modulators owing to its excellent physical properties. While conventional LN electro-optic modulators continue to be the workhorse of the modern optoelectronics, they are becoming progressively too bulky, expensive, and power-hungry to fully serve the needs of this industry. Here, we demonstrate plasmonic electro-optic directional coupler switches consisting of two closely spaced nm-thin gold nanostripes on LN substrates that guide both coupled electromagnetic modes and electrical signals that control their coupling, thereby enabling ultra-compact switching and modulation functionalities. Extreme confinement and good spatial overlap of both slow-plasmon modes and electrostatic fields created by the nanostripes allow us to achieve a 90% modulation depth with 20-μm-long switches characterized by a broadband electro-optic modulation efficiency of 0.3 V cm. Our monolithic LN plasmonic platform enables a wide range of cost-effective optical communication applications that demand μm-scale footprints, ultrafast operation and high environmental stability.

---

[1] Centre for Nano Optics, University of Southern Denmark, Campusvej 55, DK-5230 Odense M, Denmark. *email: math@mci.sdu.dk; seib@mci.sdu.dk

In the last decades, lithium niobate has become indispensable for integrated photonics as the material of choice for electro-optic modulation due to its excellent (linear and nonlinear) optical and material properties. Being advantageous over competing platforms, lithium niobate (LN) fulfills the eligibility material requirements for optical communication systems by exhibiting wide optical transparency (0.35–4.5 μm), large electro-optic coefficients ($r_{33} = 30$ pm V$^{-1}$), which are preserved at elevated temperatures due to its high Curie temperature (~1200 °C), and excellent chemical and mechanical stability resulting in long-term material reliability[1]. Leading to considerable commercial significance, the early success of LN for optoelectronic applications was driven by heterogeneous integration of metal-diffused channel optical waveguides utilized for chip-scale electro-optic modulators[2–7]. However, the weak confinement of integrated metal-diffused optical waveguides is limiting the electro-optic interaction, resulting in low electro-optic modulation efficiencies and large device footprints. Recently, monolithic integration of thin-film lithium niobate modulators[8–14] has attracted an increasing attention due to significantly higher optical confinement, leading to improvements in terms of compactness, bandwidth and energy efficiency, while still demanding relatively long, on the mm-scale, interaction lengths due to conceptual limitations in the electro-optic field overlap.

Leveraging metal nanostructures to transmit simultaneously both optical and electrical signals, with the additional attribute of extremely enhancing their accompanied local fields, promises plasmonics to become a versatile platform for exceptionally compact optoelectronic applications[15,16]. The first pioneering work[17] utilizing surface plasmon polaritons (SPPs) for electrically controlled modulation was based on thermo-optic effects induced by resistive heating in polymer materials. Though this approach facilitates only moderate switching times and relatively high power consumption, the large overlap between the electromagnetic field of the plasmonic mode and the electrically induced local change of the refractive index was opening the path to exceptionally efficient plasmonic electro-optic modulators. Following this approach, tremendous efforts have been directed towards exploring other electro-optic material platforms, which drastically improved the switching performance, including two-dimensional materials[18–21], phase-change materials[22–24], and electro-optic polymers[25–31]. These studies convincingly demonstrated the capability of plasmonics to be a potential complementary technology addressing bottleneck issues in future information technology. However, combining the attractive features of plasmonics with LN[32], to date still the preferred material platform meeting all essential performance requirements, has remained largely unexplored.

Here, we introduce a monolithic plasmonic modulator/switch configuration based on two identical gold nanostripes on LN, where the metallic structure utilized for applying external electrostatic fields inherently supports the propagation of the SPP modes, resulting in an exceptionally simple device architecture. Our approach does not require patterning, etching or milling of the LN substrate, which is particularly challenging due to its mechanical hardness and chemical stability. The antisymmetric change of the refractive index of LN due to the Pockels electro-optic effect induced by an external electric field applied across two gold nanostripes affects the optical coupling between the plasmonic modes propagating along the two nanostripes. This allows high-density integration with extraordinary efficient and broadband switching of the optical power distribution (at telecom wavelengths) between the two output ports of plasmonic directional couplers.

## Results

**Device principles**. The proposed optical switch configuration represents a plasmonic directional coupler that comprises two

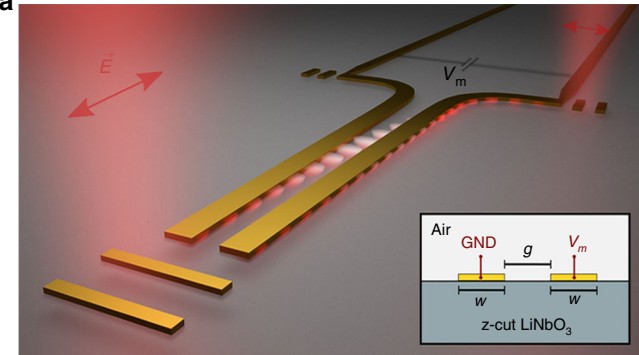

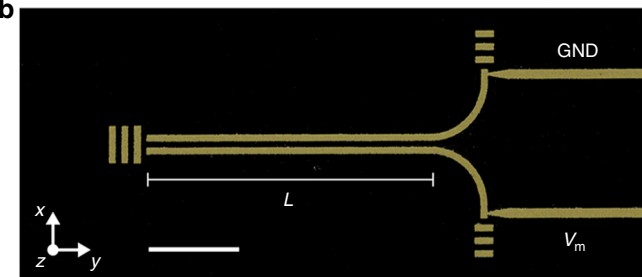

**Fig. 1 Plasmonic monolithic lithium niobate directional coupler switch. a** Conceptual image of power modulation in the plasmonic nanostripes by applying a bias voltage, which induces a refractive index change due to the electro-optic effect in the substrate. The inset shows the cross-section of the two identical parallel waveguides of width $w = 350$ nm, separated by a gap of $g = 350$ nm and placed on z-cut lithium niobate (LiNbO$_3$, LN). **b** False-colored scanning electron microscope (SEM) image of the investigated plasmonic directional coupler switch. The scale bar represents 5 μm. The optimal length $L$ of interaction is chosen to be 15.5 μm.

lithographically fabricated gold nanostripe waveguides placed on z-cut LN, which have identical dimensions of $w \times h = 350 \times 50$ nm$^2$ and are separated by the distance of 350 nm (Fig. 1). Numerical optimization of the waveguide dimensions is given in Supplementary Note 5.

The transmission line, composed of two optically coupled plasmonic strip waveguides, supports two quasi-TEM modes, i.e., the odd and even modes, which define the geometrically dependent coupling length $L_C = \pi/|\beta_{odd} - \beta_{even}| = \lambda_0/(2 \cdot |n_{odd} - n_{even}|)$ of the passive system with the mode effective indices $n_{odd}$ and $n_{even}$, respectively. An optical near-field study is conducted to verify and quantify coupling between the two SPP modes supported by the considered configuration (Supplementary Note 9 and Supplementary Movie 1). In the modulator device, light with the free-space wavelength $\lambda_0$ is fed symmetrically into the two plasmonic waveguides by positioning a diffraction-limited beam on a metallic grating coupler. A 90° bend of the individual waveguides in opposite directions is employed to separate the coupling channels and accordingly limiting the interaction length $L$ between the SPP modes supported by the nanostripe waveguides. Furthermore, the linearly polarized emission from the terminating gratings is rotated by the bend to be orthogonal to the polarization of the incident beam, thus allowing cross-polarized far-field imaging with suppression of back-reflections from the incident beam. In order to introduce an external electrostatic field in the electro-optically active substrate supporting the plasmonic coupler, the individual waveguide ends are electrically connected to signal ($V_m$) and ground (GND), respectively. The change of the refractive index induced by the Pockels effect in z-cut LN depends on the direction of the applied electric field relative to the direction of the optic axis

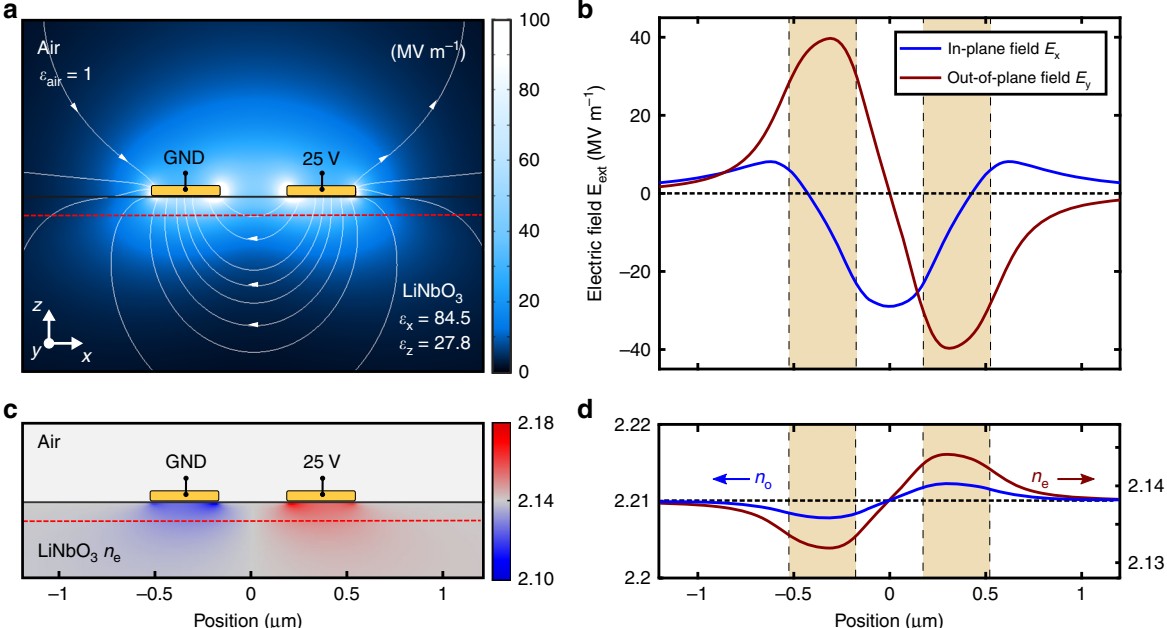

**Fig. 2 Electrostatic field simulation of the two-wire configuration. a** Color-coded electric field and its contours upon applying a bias of 25 V. The electrostatic field extends stronger along the depth-direction ($z$-axis), as compared to the isotropic case, due to the anisotropy in the relative permittivity of $z$-cut lithium niobate (LiNbO$_3$, LN). **b** In-plane ($E_x$) and out-of-plane electric-field component ($E_z$) in the LN substrate 100 nm below the surface (along the dashed red line shown in Fig. 2a). The yellow bars indicate the position of the Au wires. The in-plane field component $E_x$ shows highest amplitude in between the two electrodes, while the out-of-plane component $E_z$ exhibits highest amplitude underneath the wires with opposite sign. **c, d** Due to the linear electro-optic (Pockels) effect in the LN substrate, the extraordinary ($n_e$) and ordinary ($n_o$) refractive index underneath the left wire is decreased, while being increased by an equal amount in the right guide. Consequently, the waveguides lose their optical identity, which influences the power transfer in the directional coupler.

of the crystal. To investigate the optical response on the externally applied field, the electrostatic field distribution with the field contour lines are simulated for the two-wire configuration while one wire is grounded and the other is biased with 25 V (Fig. 2).

The field extends further along the depth-direction as compared to the isotropic case due to the anisotropic relative permittivity of LN. The amplitude of the electric field components along the $x$- and $z$- direction 100 nm below the LN/air interface (Fig. 2b) reveals an alternating electric field along the LN optical axis below the individual nanostripes.

Therefore, an opposite refractive index modification is introduced in the two waveguides, consequently eliminating the device symmetry of the directional coupler (Fig. 2c, d). Strong overlap between the plasmonic field and the modulating electrostatic field results in a significant phase change with opposite polarities in the two waveguides, thus providing efficient push-pull operation of the directional coupler modulator. An expression for the intensity modulation and switching characteristics of the device can be derived from the coupled-mode formalism[33]. Under assumption of negligible coupling in the output bends and by neglecting absorption losses, the normalized power in the weakly coupled waveguide system is given by (see Supplementary Note 1):

$$P_{1,2} = \frac{1}{2}\left[1 \pm \frac{2}{\Delta\beta_n + \Delta\beta_n^{-1}}\sin^2\left(\frac{\pi}{2}\frac{L}{L_C}\sqrt{1 + \Delta\beta_n^2}\right)\right], \quad (1)$$

where $\Delta\beta_n = [\beta_1(V_m) - \beta_2(V_m)]/[\beta_{odd}(V_m = 0) - \beta_{even}(V_m = 0)]$ is the normalized electrically induced difference in the propagation constants $\beta_1$ and $\beta_2$ of each individual waveguide mode, and $L$ is the interaction length. From Eq. (1) it follows that the unbiased device is inherently set to the linear section of the modulation transfer curve (quadrature point), offering high degree of linearity and collapse of

even-order nonlinear distortions in the modulation spectrum. It is clear that complete switching of the optical power can be reached when the device length is $L = L_C/\sqrt{2}$ and when $\Delta\beta_n = 1$ (i.e., the required voltage should cause the difference in propagation constants of each wire to be as high as the difference between mode propagation constants of the even and odd modes in the passive device). Deviations from this interaction length result in reduced maximum modulation depth and sensitivity (see Supplementary Note 1 and 2 for details). It was found that for the considered cross-section geometry of the directional coupler, this condition is fulfilled at the interaction length of $L = 15.5\,\mu m$ and thus taken as the optimum design for efficient switching in our study. Interestingly, the above conditions for the complete switching demonstrate the well-known trade-off between required voltage and device length: with less coupled waveguides, the required full-switch voltage will be smaller, but the optimal interaction length will be larger. Due to the established material platform and the simple device geometry, the fabrication procedure of the monolithic modulator device is exceedingly simple, involving only a standard electron-beam lithography step followed by thermal gold evaporation and subsequent metal lift-off (see Methods for more details). Furthermore, the switch conceptionally exhibits relaxed tolerances to geometrical variations and wavelength instabilities compared to ring resonators or Mach-Zehnder interferometers (see Supplementary Note 3–5 for details).

**Performance of the device**. Switching and modulation of the scattering signal emitted from the output gratings of the two waveguide channels in the characterized device is illustrated with far-field images shown in Fig. 3 for two different bias voltages with opposite polarity.

At zero bias, the symmetry of the device configuration results in an equal phase condition in the coupler arms and consequently

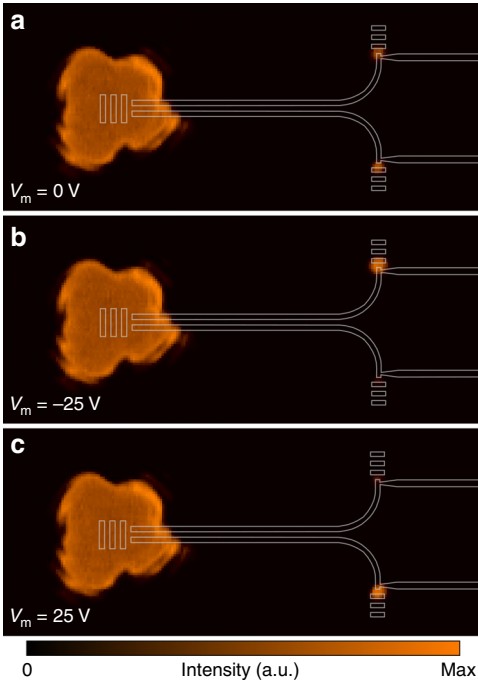

**Fig. 3 Visualization of switching and modulation behavior.** Experimental optical far-field images captured by an infrared camera at different modulation voltages. A diffraction-limited beam ($\lambda_0 = 1550$ nm) is positioned on the input grating to symmetrically feed the coupling system. The structure design is superimposed as a guide to the eye. The scattering images are superimposed with the structure design. **a** The unbiased device ($V_m = 0$ V) exhibits equal power distribution of the scattered signal emitted from the output ports, due to the symmetry of the directional coupler. **b** When a negative bias voltage ($V_m = -25$ V) is applied, an enhancement of the optical output signal of the biased waveguide arm is observed, while the optical signal at the opposite output port is decreased in intensity. **c** By changing the polarity of the applied voltage, the intensity in the two output terminals is switched.

equal power distribution in the waveguides, revealed by balanced output signals. When a bias signal is applied, an enhancement of the output signal at one port is observed, while the signal at the opposite output port is decreased. By changing the polarity of the applied voltage, the intensity in the two output terminals is switched, revealing a reversed refractive index change as expected from the linear modulation characteristics of the Pockels effect. The dynamic optical switching, induced by an alternating voltage of $V_{AC} = 25$ V at slow switching speeds, is captured by the infrared camera (Supplementary Movie 2).

For obtaining the transfer curve in Fig. 4a, the applied bias is varied and the output power modulation of the two channels is measured independently at the image plane by spatial filtering with subsequent detection by a photodiode (see the "Methods" section for more details). The experimentally measured modulation curve agrees well with the analytical predictions from the coupled mode theory (CMT), assisted by finite element method (FEM) simulations (see the "Methods" section for details). A modulation depth of 90% is achieved at sufficiently high voltages ($V \approx \pm 50$ V), without permanent breakdown of the device. This corresponds to a dual-channel intensity extinction ratio (ER) of 10 dB. Given that the full modulation depth is reached at the electrically induced phase mismatch of $\sqrt{2}\pi$, our device exhibits a drive voltage-length product of 0.3 V cm (half-wave voltage-length product of $V_\pi L = 0.21$ V · cm), which is approximately an order of magnitude smaller than that obtained with the

state-of-the-art photonic LN modulators. In fact, lower levels of the voltage-length product have only been observed with electro-optic polymers, whose very low phase-transition temperatures impede their practical deployment (see Supplementary Table 2 for a detailed comparison). The insertion loss due to the propagation loss of $\alpha = 0.35$ dB $\mu m^{-1}$ in the phase shifter section is 5.5 dB, resulting in a half-wave voltage-length-loss product of $V_\pi L \alpha = 735$ VdB. Due to non-resonant modulation characteristics and the wide optical transparency window, spectrally broadband operation of our directional coupler switch is expected. The relative modulation efficiency at the driving voltage of 15 V is measured within the wavelength range of 1280–1590 nm (Fig. 4b), normalized by that measured at the design wavelength of 1550 nm. Although the performance of the device fluctuates with the operation wavelength, a high wavelength tolerance is observed, exhibiting <2 dB modulation depth variation within all wavelength telecommunication bands.

The electro-optic frequency response of the directional coupler switch is characterized from 10 kHz to 10 GHz (Fig. 4c). The device exhibits a flat frequency response up to 2 GHz in our measurement. The observed 3-dB bandwidth of $(9 \pm 2)$ GHz is caused by a limited response bandwidth of the electrical feedline on the chip and the RF probes (see Supplementary Note 8 for details). Owing to the extremely short response time (~ fs) of the Pockels effect and the small device capacitance of only 3.6 fF, the calculated modulation cutoff frequency exceeds 800 GHz at 50 Ω resistive load ($f = 1/[2\pi RC]$), indicating potential operation at much higher modulation speeds than shown in our measurements and suggesting the energy consumption reaching levels below 1 pJ/bit (see Supplementary Note 6 for details).

## Discussion
In summary, we demonstrated a broadband directional coupler switch featuring a drive voltage-length product of 0.3 V cm at telecom wavelengths, to date the lowest value for a device based on lithium niobate electro-optic modulation. Owing to the excellent material properties such as electro-optic reliability and temperature stability, this material platform is still considered as the material-of-choice that can fulfill the demands of future optical data links, by exploring new device configurations which are significantly smaller, faster, and more efficient than current LN electro-optic modulators. The proposed directional coupler switch configuration addresses this challenge with exceptional structural simplicity by providing multiport, broadband, and effective modulation at a compact footprint and a high-speed operation. The presented proof-of-concept study of utilizing integrated plasmonic circuits in LN platforms demonstrates its enormous potential, which can pave the way towards feasible communication links, promising high-speed, broadband, and robust operation. The full-switching voltage in our configuration can be reduced by utilizing plasmonic waveguides with higher optical confinement or longer interaction length, which can eventually be interfaced with low-loss photonic waveguides or multicore fibers[26] for practical integration into long-haul on-chip optical communication systems.

## Methods
**Device fabrication.** Devices are fabricated on commercially available z-cut lithium niobate substrates. The directional coupler modulators are written by electron beam lithography (using a scanning electron microscope JOEL JSM-6490LV with an acceleration voltage of 30 keV) in spin-coated 200-nm-thick PMMA positive resist and 20-nm-thick Al layer, which serves as a metallic charge dissipation layer during the writing (electron doses varying between 200 and 250 $\mu C\ cm^{-2}$). After resist development, the directional couplers are formed by depositing a 4 nm titanium adhesion layer and a 50 nm gold layer by thermal evaporation and subsequent 12 h lift-off in acetone. To reduce electron beam writing time, macroscopic bonding pads and connecting wires are patterned beforehand on the LN chip by

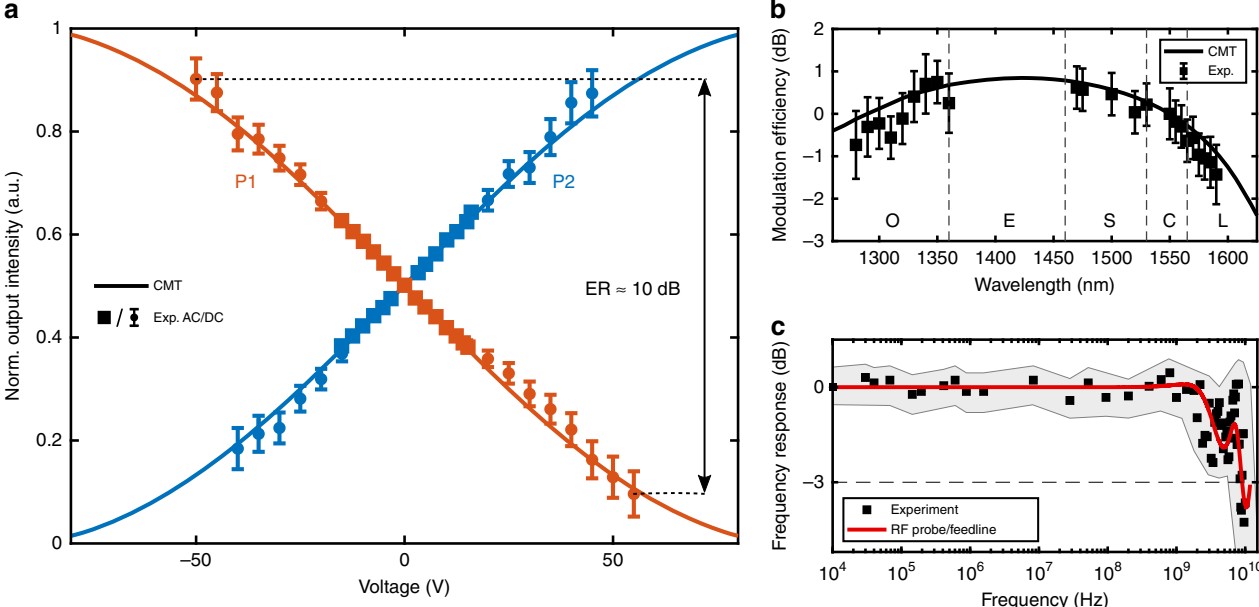

**Fig. 4 Directional coupler switch characteristics. a** Measured (symbols) and simulated (solid lines) electro-optical transfer function, showing the power exchange as a function of the applied bias voltage. Error bars represent standard deviations and are deduced from repeated measurements. For AC signals smaller than 15 V (with a frequency of 200 kHz), error bars are smaller than the symbols. **b** Wavelength dependence of the modulation efficiency (at the driving voltage of 15 V) normalized by that measured at the design wavelength of 1550 nm. Less than 2 dB modulation deterioration over all optical communication wavelength bands is observed. Error bars indicate the estimated standard deviation of the mean. **c** Measured frequency response as a function of the applied RF signal frequency, normalized with respect to the lowest applied frequency. The continuous shaded error region represents the standard deviation in the measured frequency response with a smoothed average between data points. The response in the measured frequency range is only restricted by the bandwidth limitations of the RF probe and the electrical feedline (red line).

optical lithography, metal deposition (5 nm Ti/150 nm Au), and lift-off. Lithographic overlay is ensured by the mix-and-match approach.

**Experimental electro-optical characterization**. A collimated and linearly polarized laser beam from a tunable IR laser is focused by a high numerical aperture objective (Olympus MPlan100xIR, NA = 0.95). The focused beam is positioned symmetrically on the grating coupler for symmetric feeding of the two arms of the directional coupler. The output optical signal is collected by the same objective. The cross-polarized configuration is used to suppress the back-reflection and scattering from the incident beam. This provides an improved signal-to-noise ratio for the optical signal scattered from the output gratings, which is subsequently detected by the IR camera or a high-speed photodiode. The integration time of the camera is adjusted to give a reasonable contrast for the output signal, leading to an over-saturation and increased size of the incident back-reflected beam spot. By applying electrical probes on the electrode pads, one directional coupler arm is grounded while the other is biased with an AC/DC signal. For measuring the modulation transfer curve, the wavelength dependence and the frequency response of the device, the scattering signal coming from one port of the directional coupler is spatially filtered and detected by a high-speed photodetector. The frequency response measurements are calibrated by the response reference of the signal generator and the photoreceiver. A detailed description of the experimental set-up can be found in Supplementary Note 7 and 8.

**Numerical modeling**. FEM simulations are performed using a commercially available software (Comsol Multiphysics 5.2a). For the electrostatic simulations, the device cross-section is modeled with the unclamped static relative permittivity tensor of LN taken from Jazbinsek et al.[34] ($\varepsilon_{xx} = \varepsilon_{yy} = 27.8$, $\varepsilon_{zz} = 84.5$), while the relative permittivity of air is set to be $\varepsilon_{air} = 1$ and the boundaries of the two gold nanostripes are set to ground and $V_{bias}$ potential, respectively. The calculated electric-field distribution is utilized to determine the modification of the refractive index in the LN substrate by using the electro-optic Pockels coefficients from Jazbinsek et.al.[34] (restricting to the largest diagonal terms, i.e., $\Delta n_{ii} = -0.5 r_{iiz} n_{ii}^3 E_z$, with $r_{xxz} = r_{yyz} = 10.12$ pm/V and $r_{zzz} = 31.45$ pm V$^{-1}$), which are considered to be non-dispersive over the investigated wavelength range[35]. For the optical simulation, the modified distribution of the refractive index of LN is fed into the mode solver, while the unmodified refractive indices from Au and LN were taken from Johnson and Christy[36] and Zelmon et al.[37] ($n_{xx} = n_{yy} = n_o = 2.211$, $n_{zz} = n_e = 2.138$ at $\lambda_0 = 1550$ nm). Scattering-boundary conditions in combination with a perfectly matched layer are applied to calculate the modification $\beta_1 - \beta_2$ in the propagation constants of each individual waveguide as a function of the applied

voltage, which is used for describing the intensity modulation using the coupled-mode formalism (see Supplementary Note 1 for details). The optical simulation of two parallel waveguides and unmodified LN refractive index was used to calculate the propagation constants of even and odd modes ($\beta_{even}$ and $\beta_{odd}$) and, eventually, the coupling length $L_C$.

## Data availability
All data that support the findings of this study are available from the corresponding authors upon reasonable request.

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

## Acknowledgements

The authors acknowledge funding support from the European Research Council (PLAQNAP, Grant 341054), the University of Southern Denmark (SDU2020 funding) and the VILLUM FONDEN (Grant 16498). C.W. acknowledges funding from a MULTIPLY fellowship under the MarieSkłodowska-Curie COFUND Action (grant agreement No. 713694). We also thank Martin Houmann Thygesen, Jesper Nielsen and Morten Nymand from SDU Electrical Engineering for providing equipment for the high-speed measurements. We acknowledge Marcus Fred Hufe for supporting the near-field measurements and Joel Cox for constructive feedback on the manuscript.

## Author contributions

S.I.B. and M.T. conceived the experiment and geometry of the modulators. M.T. set up the experiment, performed the device fabrication, experimental characterization and simulations of the device. V.Z. and C.W. contributed in the device characterization and simulations. M.T. wrote the manuscript with contributions from all other authors. S.I.B. supervised the project.

## Competing interests

The authors declare no competing interests.
