## [Peer Review File · Nature Communications]

Reviewers' Comments:

Reviewer #1:

Remarks to the Author:

The manuscript "Plasmonic monolithic lithium niobate directional coupler switches" by Thomaschewski et al., demonstrated very low $V_{\pi}L$ value of 0.3 V-cm by combining the lithium niobate and plasmonic waveguide.

Although the manuscript has shown the best performance in terms of $V_{\pi}L$, I do not think its quality meets Nature Communication's high standard. In the following, I give the reason for the such evaluation is given.

Major issues:

- Most importantly, the overall quantity and quality of the measured data is rather moderate. Although figure 3 clearly shows how devices operate, it does not carry much information about device performance. Therefore, only Figure 4 has proper measured data but the quality of measurement is not as good as other works in similar platform, such as Refs [1 - 3]. For example, electro-optic response S_{21} in all three references go above 10 GHz, however, this manuscript measures up to 100 MHz with comparatively few data points. In addition, eye-diagram in all three references do exist in the manuscript.
- One of the well-known problems of plasmonic devices is the insertion loss caused by absorption of metal. In that sense, information about its loss should be given in the main text, rather than in the supplementary material. It would have been much better if either $\alpha V_{\pi}L$ (information in the Supplementary Table 2) or αV_{π}^2L should be given in the main text. Furthermore, implication of the loss for practical deployment, trade-off between loss and energy or potential route for improvement could have added more values.

Minor issues:

- In all subfigures of Fig. 4, unit of all data are somewhat normalized. But there is not enough explanation of normalization is given. Especially, there are data points above 0dB after normalization.

References

1. S. Abel, T. Stöferle, C. Marchiori, C. Rossel, M. D. Rossell, R. Erni, D. Caimi, M. Sousa, A. Chelnokov, B. J. Offrein, and J. Fompeyrine, Nature Communications 4, (2013).
2. K. Alexander, J. P. George, J. Verbist, K. Neyts, B. Kuyken, D. Van Thourhout, and J. Beeckman, Nature Communications 9, (2018).
3. C. Wang, M. Zhang, X. Chen, M. Bertrand, A. Shams-Ansari, S. Chandrasekhar, P. Winzer, and M. Lončar, Nature 562, 101 (2018).

Reviewer #2:

Remarks to the Author:

The manuscript reports on an ultra-compact lithium niobate electrooptic switching directional coupler, acting as a demonstration of a nanophotonic platform technology incorporating plasmonic waveguides. With a record low Voltage-length product figure of merit and low capacitance, this is an important result in the context of high density photonic integration.

The manuscript reports on an ultra-compact lithium niobate electrooptic switching directional coupler, acting as a demonstration of a nanophotonic platform technology incorporating plasmonic waveguides. With a combination of low voltage-length product figure of merit, fast nonlinear response and low capacitance, this is an important result in the context of enabling high density photonic integration.

While plasmonic active devices have been studied extensively in recent years, and lithium niobate remains the essential material of choice for electrooptic modulators, the manuscript is significant for combining the two into a platform technology and for demonstrating a highly compact device geometry that leverages plasmonic field confinement. The report may be of broad interest to any readers with an interest in breakthrough technologies in optical communications, as well as to specialists in optical materials related disciplines.

The data and their presentation appear to be of high quality. The supplemental notes contain detailed information on all the key aspects of the experiment, including the details that would be relevant to reproduction. The conclusions appear to be reliable, and the paper appropriately cites relevant references.

Supplemental Table 2 is an excellent comparison of the newly reported device in context with other state of the art electro-optic modulators. (This table is cited twice in the main text section "Performance of the Device," and the discussion might be clearer for readers if the table itself appears in the main text.)

While the manuscript overall is excellent, and the technical writing is good throughout most of the document, the abstract and introduction are not written with great clarity. There are a moderate number of minor grammatical or typographical errors, but in particular the 2nd sentence of the abstract is very unclear and needs to be re-edited.

CORRECTIONS MADE IN REPLY TO REVIEWER 1'S COMMENTS

COMMENT 1

REVIEWER: Most importantly, the overall quantity and quality of the measured data is rather moderate. Although figure 3 clearly shows how devices operate, it does not carry much information about device performance. Therefore, only Figure 4 has proper measured data but the quality of measurement is not as good as other works in similar platform, such as Refs [1 - 3]. For example, electro-optic response S21 in all three references go above 10 GHz, however, this manuscript measures up to 100 MHz with comparatively few data points. In addition, eye diagram in all three references do exist in the manuscript.

References

1. S. Abel, T. Stöferle, C. Marchiori, C. Rossel, M. D. Rossell, R. Erni, D. Caimi, M. Sousa, A. Chelnokov, B. J. Offrein, and J. Fompeyrine, *Nature Communications* 4, (2013).
2. K. Alexander, J. P. George, J. Verbist, K. Neyts, B. Kuyken, D. Van Thourhout, and J. Beeckman, *Nature Communications* 9, (2018).
3. C. Wang, M. Zhang, X. Chen, M. Bertrand, A. Shams-Ansari, S. Chandrasekhar, P. Winzer, and M. Lončar, *Nature* 562, 101 (2018).

OUR REPLY: We would like to thank the reviewer for this comment, which raised an important point and encouraged us to substantially improve the quality of our manuscript. We have significantly revised our experiment with respect to the device characterization to address this comment by further improving the quantity and quality of the measured data.

This includes a demonstration of our device functioning over a broader wavelength range of 1280 nm - 1590 nm (instead of 1550 nm - 1630 nm in our previous submission). These measurements thus emphasize an essential advantage of LN over other material platforms (as, for example, those used in the above Refs. 1 and 2) and the proposed non-resonant configuration over resonant ones (as, for example, that used by C. Haffner *et al.*, *Nature* **556**, 483-486, 2018), of being suitable to switch and modulate (and transmit) optical signals in an exceptionally broad wavelength range. In fact, less than 2 dB performance deterioration over all optical communication wavelength bands was observed.

Furthermore, by significantly upgrading our experimental set-up and improving RF impedance matching, the electro-optic response of our total device (directional coupler + connecting electric wires) is measured up to 10 GHz, thus presenting a **100-fold improvement** over our previously reported response up to 100 MHz. A response bandwidth of approx. 9 GHz (3-dB bandwidth) was measured, which was identified to be restricted by the coplanar electrical feedline used to transmit the electrical signal on the chip and the limited bandwidth of the RF probes. This was analyzed in an additional section in the Supplementary material (section VIII.). By considering the mentioned bandwidth parasites in our experimental set-up, we do not observe any bandwidth limitation from the plasmonic device itself in the measured frequency range, as expected from the low device's capacitance of 3.6 fF.

Altogether, we believe that these radical improvements in our manuscript now meets the high standards of *Nature Communications*.

OUR MODIFICATION: We replaced Figure 4 in our manuscript by the Figure shown below, now including the extended measurement:

Figure 4 | Directional coupler switch characteristics. **a** Measured (symbols) and simulated (solid lines) electro-optical transfer function, showing the power exchange as a function of the applied bias voltage. **b** Wavelength dependence of the modulation efficiency (at the driving voltage of 15 V) normalized by that measured at the design wavelength of 1550 nm. Less than 2 dB modulation deterioration over all optical communication wavelength bands is observed. **c** Measured frequency response as a function of the applied RF signal frequency with its uncertainty (shaded area), normalized with respect to the lowest applied frequency. The response in the measured frequency range is only restricted by the bandwidth limitations of the RF probe and the electrical feedline (red line).

Besides the updated description of the setup (Supplementary Section VII), an additional section in the supplementary information was added (Supplementary Section VIII), which explains the observed frequency response limitation due to external parasitic bandwidth restrictions:

RF characterization

The limited sensitivity of the photodetector (Thorlabs PDA8GS) and the fact that we did not include an optical amplifier in the detection path resulted in an RF transfer function of the overall system of as low as -140 dB. We did not have access to a vector network analyzer of the required bandwidth and dynamic range. Instead, we measured the RF response by exciting the modulator with a microwave synthesizer (HP8672S, excitation level approx +10 dBm) and

detecting the photocurrent with a separate RF spectrum analyzer (Rohde&Schwarz FSM, noise floor approx. -150 dBm at 6 Hz resolution bandwidth).

The plasmonic modulator was placed at the end of a coplanar waveguide composed of a 150 nm thick gold wire on the LiNbO₃ substrate with a characteristic impedance of 50 Ω +/- 20%, a length of approx. 5 mm and an Ohmic resistance of 20-30 Ω . This waveguide was terminated by two thin gold wires (50nm thick, designed to be 100 Ω each) fabricated together with the modulator. The termination wire resistance was subject to considerable variation from sample to sample leading to tolerable ripple in the RF-response.

On the opposite end, the coplanar waveguide was contacted by a low-profile RF-probe that was made in-house specifically to fit in the working distance of the microscope objective. It involved three contact tips, which formed a 2 mm short poorly matched coplanar waveguide (approx. 200 Ω characteristic impedance). Finally, the optical signal was detected with a commercial photoreceiver module connected directly to the spectrum analyzer. The detector is specified with a bandwidth of 9.5 GHz and according to the vendor has a first-order low-pass response.

The low RF signal at the receiver led to significant problems with direct crosstalk due to leakage from RF cables and through power cords. Because of the strong background signal at the microwave stimulus frequency and the sidebands at the harmonics of the power grid frequency, we modulated the optical incident power with a chopper at 75 Hz introducing RF-sidebands that were an unambiguous signature of the optical modulator.

We calibrated the frequency dependence of the stimulus power level caused by variations in the synthesizer and the cablework by replacing the RF contacts with a bolometric power detector (HP8481B). Furthermore, we subtracted the typical response of the photodetector (first-order low-pass pole at 9.5 GHz) from the total signal. Since we did not have access to a vector network analyzer or a reflectometer with the necessary bandwidth, we could not calibrate out the RF response of the contact pads and the on-chip waveguide. As an alternative, we simulated it using designed (characteristic impedance of the waveguide and the contact tips) and measured parameters (Ohmic resistance of the waveguide and the termination resistor). The resulting RF-response is shown in Fig. S11 alongside an equivalent schematic. We find that this RF model is in excellent agreement with the measured data (both shown in Fig. 4c). We stress that the model

was not fitted to the measured data but was based on unrelated measurements. Therefore, we are confident that the apparent drop in modulated signal at higher frequencies is due to imperfections in the RF contacts and feedline. We do not observe any bandwidth limitation from the plasmonic modulator itself as expected from our earlier estimates based on the device's capacitance.

Figure S11. The resulting RF response of the designed (characteristic impedance of the waveguide and the contact tips) and measured parameters (Ohmic resistance of the waveguide and the termination resistor) in our experiment, alongside an equivalent schematic (inset).

COMMENT 2

REVIEWER: One of the well-known problems of plasmonic devices is the insertion loss caused by absorption of metal. In that sense, information about its loss should be given in the main text, rather than in the supplementary material. It would have been much better if either $\alpha V\pi L$ (information in the Supplementary Table 2) or $\alpha V\pi^2 L$ should be given in the main text. Furthermore, implication of the loss for practical deployment, trade-off between loss and energy or potential route for improvement could have added more values.

OUR REPLY: We thank for the suggestion of giving more information about the loss in our plasmonic directional coupler switch. We agree with the given remark, indeed the plasmonic losses present a major hurdle in applications in which the impact on insertion losses are crucial, while bulkier device footprints and less-denser integration are being tolerated. To analyze the trade-off between loss and modulation efficiency including the potential improvements in our device configuration, we conducted a parametric study of the influence of the structure's dimensions on the half-wave voltage-length product $V\pi L$, the propagation loss α and the half-wave voltage-length-loss product $V\pi L\alpha$. Although our device requires a phase mismatch of $\sqrt{2}\pi$ for full switching, it is reasonable to compare the figures of merit in a more general context for an electro-optically induced phase mismatch of π (half-wave voltage-length product), which generalizes the modulator performance for potential other device platforms (Mach-Zehnder

modulators, ring-resonators). The results for a fixed electrode separation distance of 350 nm and varying waveguide width and height are summarized in Figure S7 (see below in our modifications). It is shown that the half-wave voltage-length product $V_{\pi}L$ can be reduced to below 0.2 V·cm by reducing the waveguide dimensions and thus tightening optical confinement, causing a stronger light-matter interaction. Contrarily, as the mode confinement becomes tighter, propagation losses are increasing in smaller waveguides (Figure S7b). This regime might be of interest for resonant device platforms where attenuation is desired i.e. in the device's off state, to electro-optically switch the resonant response and selectively bypassing the plasmonic loss in the device's on state. (Haffner, C. *et al.* Low-loss plasmon-assisted electro-optic modulator. *Nature* **556**, 483–486, 2018). For non-resonant waveguide-based device platforms in which the ohmic loss is unwanted, an optimal trade-off between electro-optic efficiency (Figure S7a) and loss (Figure S7b) must be found. Therefore, by multiplying the figures of merit, we defined the voltage-length-loss product $V_{\pi}L\alpha$. Since the loss is decreasing faster with larger waveguide dimensions than $V_{\pi}L$ increases, we observe a steady decrease of the loss-voltage-length product for larger waveguide dimensions. In fact, for each waveguide width one can find an optimal waveguide height (i.e., the gold stripe thickness), in which best performance is found (shown by the red line in Figure S7c). The actual device dimensions used in our experiment are thus chosen to be close to this optimum. In our directional coupler modulator, other relevant parameters such as the coupling length dictate the device performance (Figure S8a). As described in *Supplementary Note 1*, the optimum device length is $L = L_c\sqrt{2}$. By multiplying the optimum device length with the propagation loss α , we can determine the insertion loss for different waveguide geometries (Figure S8b). In our proof of concept study, we targeted a small voltage-length product with an acceptable optical insertion loss which is kept below 6 dB, thus exhibiting comparable insertion losses as commercially available electro-optic modulators (*Thorlabs, Lithium Niobate Modulators*, <https://www.thorlabs.com>; *Eospace 2017 Advanced Products*. <http://eospace.com/pdf/EOSPACebriefProductInfo2017.pdf>). This condition is achieved at waveguide dimensions of $350 \times 50 \text{ nm}^2$.

Overall, including the insertion loss in the figure of merit for our plasmonic modulator would favor devices with larger footprints and half-wave voltage-length products, a trend that might not universally be accepted, especially in the presence of other losses (for example, coupling losses) in the device. Therefore, in response to Comment 2, we describe the efficiency-loss tradeoff in our design considerations with an additional section in the supplementary information, while explicitly mentioning the insertion loss of our device in the main text.

OUR MODIFICATION: Corresponding changes to the manuscript (page 10):

“In fact, lower levels of the voltage-length product have only been observed with electro-optic polymers, whose very low phase-transition temperatures impede their practical deployment (see Supplementary Table 2 for a detailed comparison). **The insertion loss due to the propagation loss of $\alpha = 0.35 \text{ dB}/\mu\text{m}$ in the phase shifter section is 5.5 dB, resulting in a half-wave voltage-length-loss product of $V_{\pi}L\alpha = 735 \text{ VdB}$. Due to non-resonant modulation...**”

Corresponding changes to the Supplementary information:

We added a comprehensive parametric study of the modulators FOM as a function of the device dimensions in the supplementary information:

V. Design optimization for plasmonic dual-channel phase modulators and directional coupler switches

In this section, a numerical optimization of the cross-section design is presented first in the general context of a dual-channel phase modulator. Since its practical implication is not restricted to the directional coupler switch (other device platforms like Mach-Zehnder interferometers or resonators are possible), we investigate the influence on the cross-section geometry on the half-wave voltage-length product for a phase shift of π between the two waveguides, the plasmonic losses and its interplay as voltage-length-loss product (FOM = $V_\pi L \alpha$). For simplicity, the separation distance between the two wires is fixed at 350 nm as in our experimentally realized device. By reducing the waveguide dimension, the voltage-length product $V_\pi L$ can be reduced to below 0.2 V·cm (Figure S7a). This is due to the tight optical confinement in smaller waveguides which enhances the light-matter interaction of the propagating mode (i.e. the mode is more confined close to the stripe where the refractive index change is largest). Contrarily, as the mode confinement becomes tighter, propagation losses are increasing for smaller waveguide dimensions (Figure S7b). For waveguide-based device platforms in which the total optical losses should be kept reasonably small, an optimal trade-off between electro-optic efficiency (Figure S7a) and loss (Figure S7b) must be found. Therefore, by multiplying the figures of merit, we defined the voltage-length-loss product $V_\pi L \alpha$ (Fig. S7c). Since the losses decrease faster than $V_\pi L$ increases with larger waveguide dimensions, we observe a steady decrease of the voltage-length-loss product for larger waveguide dimensions. Nevertheless, for each waveguide width one can find an optimal waveguide thickness in which best device performance regarding $V_\pi L \alpha$ is found, indicated as red line in Fig. S7c. The used waveguide geometry in our experiment (marked as black diamond in Fig. S7) is close to this optimal condition.

In our directional coupler modulator, another relevant parameter that dictates the device performance is the coupling length L_c (Figure S7d). As described in Supplementary Note I, the optimal interaction length is $L = L_c \sqrt{2}$. By multiplying the optimum interaction length with the propagation loss α , one can determine the insertion loss of the directional coupler (Figure S7e). In our study, we target a small voltage-length product with an acceptable optical insertion loss which is kept below 6 dB, thus exhibiting comparable insertion losses as in commercially available electro-optic modulators¹⁰. By keeping the waveguide thickness at 50 nm and further

decreasing the waveguide width, the voltage-length product can be reduced while the insertion loss (due to plasmonic propagation) is kept below 6 dB, since the optimum interaction length scales with the reduced coupling length. However, this only provides minor performance improvements and introduces potential performance fluctuations due to fabrication imperfections (i.e. stronger thickness dependence of $V_\pi L$ and α at smaller waveguide width). Furthermore, the maximum extinction ratio is limited in lossy (plasmonic) directional couplers, when $L_c \gtrsim L_{prop}$, due to the difference in propagations losses in even and odd mode. The maximum extinction ratio in conventional passive directional-coupler can be defined by¹¹

$$ER_{\max} = \max(P_1/P_2) = \{\tanh(\text{Im}[\Delta n]k_0 L_c)\}^{-2}$$

where the difference in the effective mode indices of the odd and even mode is $\Delta n = n_{\text{odd}} - n_{\text{even}}$, k_0 is the free-space wave vector and L_c is the coupling length. Our passive directional coupler device is expected to have a maximum extinction ratio of more than 20 dB (Fig. S7f), which is well below the electro-optically induced extinction ratio of 10 dB in the active device.

Figure S7. (a-c) Design optimization for the plasmonic phase modulator. Figures of merit (**a** Half-wave voltage-length product $V_\pi L$, **b** propagation loss α , **c** voltage-length-loss product $V_\pi L\alpha$) are presented as a function of the waveguide dimensions with fixed waveguide spacing of 350 nm. The red line in **c** indicates the optimal width-thickness ratio where the smallest voltage-length-loss product can be found. **(d-f) Parametric study of the**

plasmonic directional coupler. d Coupling length, **e** insertion loss (IL) and **f** maximum extinction ratio (ER) as a function waveguide width and thickness of a directional coupler with fixed separation distance of 350 nm. The black diamond shows the design parameters used in this work ($350 \times 50 \text{ nm}^2$).

COMMENT 3

REVIEWER: In all subfigures of Fig. 4, unit of all data are somewhat normalized. But there is not enough explanation of normalization is given. Especially, there are data points above 0dB after normalization.

OUR REPLY: We thank the reviewer for the suggestion of giving more information about the normalization procedure in the subfigures of Figure 4. According to our response of comment 1, we now clarified the normalization (0-dB level) in Figure 4b, to be normalized by the design wavelength of 1550 nm and in Figure 4c by the response value at the lowest measured modulation frequency.

OUR MODIFICATION: Please see modifications made corresponding to comment 1.

CORRECTIONS MADE IN REPLY TO REVIEWER 2'S COMMENTS

COMMENT 1

REVIEWER: The manuscript reports on an ultra-compact lithium niobate electrooptic switching directional coupler, acting as a demonstration of a nanophotonic platform technology incorporating plasmonic waveguides. With a record low Voltage-length product figure of merit and low capacitance, this is an important result in the context of high density photonic integration.

The manuscript reports on an ultra-compact lithium niobate electrooptic switching directional coupler, acting as a demonstration of a nanophotonic platform technology incorporating plasmonic waveguides. With a combination of low voltage-length product figure of merit, fast nonlinear response and low capacitance, this is an important result in the context of enabling high density photonic integration.

While plasmonic active devices have been studied extensively in recent years, and lithium niobate remains the essential material of choice for electrooptic modulators, the manuscript is significant for combining the two into a platform technology and for demonstrating a highly compact device geometry that leverages plasmonic field confinement. The report may be of broad interest to any readers with an interest in breakthrough technologies in optical communications, as well as to specialists in optical materials related disciplines.

The data and their presentation appear to be of high quality. The supplemental notes contain detailed information on all the key aspects of the experiment, including the details that would be

relevant to reproduction. The conclusions appear to be reliable, and the paper appropriately cites relevant references.

Supplemental Table 2 is an excellent comparison of the newly reported device in context with other state of the art electro-optic modulators. (This table is cited twice in the main text section “Performance of the Device,” and the discussion might be clearer for readers if the table itself appears in the main text.)

While the manuscript overall is excellent, and the technical writing is good throughout most of the document, the abstract and introduction are not written with great clarity. There are a moderate number of minor grammatical or typographical errors, but in particular the 2nd sentence of the abstract is very unclear and needs to be re-edited.

OUR REPLY: We thank for the reviewer for the encouraging comment that the “data and their presentation appear to be of high quality” and that the manuscript “overall is excellent”.

We understand the reviewer’s suggestion of inserting Supplementary Table 2 in the main text for clarifying the excellent device performance regarding the low voltage-length product and good thermal stability compared to other works. However, we believe a full comparison is subject of considering a variety of figures-of-merits with its different importance for a particular application. Describing that in the main text without provoking misinterpretation by the reader would require a much more comprehensive comparison (e.g. in a review paper) than the table can provide and eventually extends beyond the manuscript’s scope. Thus, we decided to keep the table in the Supplementary information.

We are very grateful that the reviewer pointed out some weakness in the technical writing. To address the reviewer’s concerns, we have substantially revised the entire manuscript. Particularly, we rewrote the abstract and the introduction, by correcting the grammatical and typographical errors and by improving clarity in our writing. The modifications are marked below.

OUR MODIFICATION:

Abstract. From the onset of high-speed optical communications, lithium niobate (LN) has been the material of choice for electro-optic modulators owing to its large electro-optic response, wide transparent window, excellent thermal stability, mechanical and chemical resistivity resulting in the long-term material reliability. While conventional LN electro-optic modulators continue to be the workhorse of the modern optoelectronics, they are becoming progressively too bulky, expensive, and power-hungry to fully serve the needs of this industry, which is rapidly progressing towards highly integrated, cost-effective and energy efficient components and circuits. Recently developed monolithic LN nanophotonic platforms enable the realization of

electro-optic modulators that are significantly improved in terms of compactness, bandwidth and energy efficiency, while still demanding relatively long (on the mm-scale) interaction lengths. Here we successfully address this challenge and demonstrate plasmonic electro-optic directional coupler switches consisting of two closely spaced nm-thin gold nanostripes monolithically fabricated on LN substrates that guide both coupled electromagnetic modes and electrical signals influencing their coupling and thereby enabling ultra-compact switching and modulation functionalities. The extreme confinement of both slow-plasmon modes and electrostatic fields created by the two nanostripes along with their nearly perfect spatial overlap allowed us to achieve a 90% modulation depth with 20- μm -long switches characterized by a broadband electro-optic modulation efficiency of 0.3 V $\cdot\text{cm}$ at telecom wavelengths and potential energy consumption of < 1 pJ/bit. Our monolithic LN plasmonic platform enables ultra-dense integration of high-performance active photonic components, enabling a wide range of cost-effective optical communication applications that demand μm -scale footprints, ultrafast operation, robust design, and high environmental stability.

Introduction

In the last decades, lithium niobate has become indispensable for integrated photonics as the material of choice for electro-optic modulation due to its excellent (linear and nonlinear) optical and material properties. Being advantageous over competing platforms, lithium niobate (LN) fulfills the eligibility material requirements for optical communication systems by exhibiting wide optical transparency (0.35–4.5 μm), large electro-optic coefficients ($r_{33} = 30$ pm/V), which are preserved at elevated temperatures due to its high Curie temperature ($\sim 1200^\circ\text{C}$), and excellent chemical and mechanical stability¹. Leading to considerable commercial significance, the early success of LN for optoelectronic applications was driven by heterogeneous integration

of metal-diffused channel optical waveguides utilized for chip-scale electro-optic modulators²⁻⁷. However, the weak confinement of integrated metal-diffused optical waveguides is limiting the electro-optic interaction, resulting in low electro-optic modulation efficiencies and large device footprints. Recently, monolithic integration of thin film lithium niobate modulators⁸⁻¹⁴ has attracted an increasing attention due to significantly higher optical confinement, leading to improvements in terms of compactness, bandwidth and energy efficiency, while still demanding relatively long, on the mm-scale, interaction lengths due to conceptual limitations in the electro-optic field overlap.

Leveraging metal nanostructures to transmit simultaneously both optical and electrical signals, with the additional attribute of extremely enhancing their accompanied local fields, promises plasmonics to become a versatile platform for exceptionally compact optoelectronic applications^{15,16}. The first pioneering work¹⁷ utilizing surface plasmon polaritons (SPPs) for electrically controlled modulation was based on thermo-optic effects induced by resistive heating in polymer materials. Though this approach facilitates only moderate switching times and relatively high power consumption, the large overlap between the electromagnetic field of the plasmonic mode and the electrically induced local change of the refractive index was opening the path to exceptionally efficient plasmonic electro-optic modulators. Following this approach, tremendous efforts have been directed towards exploring other electro-optic material platforms, which drastically improved the switching performance, including two-dimensional materials¹⁸⁻²¹, phase-change materials²²⁻²⁴ and electro-optic polymers²⁵⁻³¹. These studies convincingly demonstrated the capability of plasmonics to be a potential complementary technology addressing bottleneck issues in future information technology. However, combining the

attractive features of plasmonics with LN³², to date still the preferred material platform meeting all essential performance requirements, has remained largely unexplored.

Here, we introduce a monolithic plasmonic modulator/switch configuration based on two identical gold nanostripes on LN, where the metallic structure utilized for applying external electrostatic fields inherently supports the propagation of the surface plasmon polariton (SPP) modes, resulting in an exceptionally simple device architecture. Our approach does not require patterning, etching or milling of the LN substrate, which is particularly challenging due to its mechanical hardness and chemical stability. The antisymmetric change of the refractive index of LN due to the Pockels electro-optic effect induced by an external electric field applied across two gold nanostripes affects the optical coupling between the plasmonic modes propagating along the two nanostripes. This allows high-density integration with extraordinary efficient and broadband switching of the optical power distribution (at telecom wavelengths) between the two output ports of plasmonic directional couplers.

OTHER MODIFICATION:

- Figure 2 has been revised by correcting the colorbar in subfigure 2a. Also, an xyz axis orientation is added. Furthermore, subfigure 2d was added, in order to show the absolute extraordinary and ordinary refractive index changes 100 nm below the LN/Air interface. Accordingly, the figure caption was rewritten.
- Although representing the same physics (response of the intensity distribution on the electro-optically induced phase-mismatch), Equation 1 is rewritten to be easier to interpret by the reader and is now fully derived in the supplementary information (Supplementary Note I).
- The main text was thoroughly rewritten. All changes made to the text are marked in an additional file in red color.
- Section I (Coupled mode formalism) of the Supplementary Information is thoroughly revised. Equation 1 from the main text is here now fully derived.
- Section II (Numerical electrostatic and mode analysis) is added to the Supplementary Information. Here, the numerical procedure to calculate the electro-optic response of the directional coupler with the context of the coupled mode theory (CMT) is described.
- The influence of the free-space wavelength on the plasmonic switch performance is now in detail described in a new section III (Influence of free-space wavelength) in the Supplementary Information.
- The description of the experimental set-up (section VII in the Supplementary Information) is rewritten, since the RF measurement required a drastic update which is now described in this section.
- The RF limitations due to parasitic bandwidth restrictions are formulated in an additional section in the Supplementary Information (section VIII.).
- Our operation voltage-length product of 0.3 V·cm in Supplementary Table 2 was replaced by the half-wave voltage-length product $V_{\pi} \cdot L = 0.21$ V·cm for having a consistent comparison with other works where the half-wave voltage-length product $V_{\pi} \cdot L$ was stated.
- References in the main text and the Supplementary Information are added/updated

Reviewers' Comments:

Reviewer #1:

Remarks to the Author:

Authors have addressed points raised by myself and performed due diligence to clarify the device performance. Given the difficulty of plasmonic device characterization, it seems that information in figure 4 is enough. Eye diagram would have been the best demonstration, however, given the difficulty of plasmonic device characterization, it is understandable that it is replaced by measurements in figure 4.

Furthermore, manuscript now clarifies the loss metric as well in response to comments by reviewer 2. This piece of information will be valuable to the research community.

Despite loss issue, as far as I know, energy figure of merit ($V_{pi} * L$) seems to be the world-record. With clear demonstration of device performance by the current manuscript, it would be worth publishing in Nature Communications.

Reviewer #2:

Remarks to the Author:

The manuscript, while of good quality originally, has been significantly improved under revision. The discussion throughout the paper is clearer. Additionally, the authors' efforts to upgrade their device characterization method provide a more impactful result, by directly measuring the device performance at much higher RF frequencies and at a broader range of optical wavelengths. The manuscript meets a high standard for quality and significance and is suitable for publication in its current form.

CORRECTIONS MADE IN REPLY TO REVIEWER 1'S COMMENTS

COMMENT 1

REVIEWER: Reviewer #1 (Remarks to the Author):

Authors have addressed points raised by myself and performed due diligence to clarify the device performance. Given the difficulty of plasmonic device characterization, it seems that information in figure 4 is enough. Eye diagram would have been the best demonstration, however, given the difficulty of plasmonic device characterization, it is understandable that it is replaced by measurements in figure 4.

Furthermore, manuscript now clarifies the loss metric as well in response to comments by reviewer 2. This piece of information will be valuable to the research community.

Despite loss issue, as far as I know, energy figure of merit ($V_{pi} * L$) seems to be the world-record. With clear demonstration of device performance by the current manuscript, it would be worth publishing in Nature Communications.

OUR REPLY: We thank the reviewer for his/her recommendation of our work.

OUR MODIFICATION: No modifications needed.

CORRECTIONS MADE IN REPLY TO REVIEWER 2'S COMMENTS

COMMENT 1

REVIEWER: The manuscript, while of good quality originally, has been significantly improved under revision. The discussion throughout the paper is clearer. Additionally, the authors' efforts to upgrade their device characterization method provide a more impactful result, by directly measuring the device performance at much higher RF frequencies and at a broader range of optical wavelengths. The manuscript meets a high standard for quality and significance and is suitable for publication in its current form.

OUR REPLY: We thank the reviewer for his/her recommendation of our work.

OUR MODIFICATION: No modifications needed.